# Anticipatory changes in British household purchases of soft drinks associated with the announcement of the Soft Drinks Industry Levy: A controlled interrupted time series analysis

Nina T. Rogers[1]*, David Pell[1], Tarra L. Penney[1,2], Oliver Mytton[1], Adam Briggs[3,4], Steven Cummins[5], Mike Rayner[6], Harry Rutter[7], Peter Scarborough[3,6], Stephen J. Sharp[1], Richard D. Smith[8], Martin White[1], Jean Adams[1]

1 MRC Epidemiology Unit, University of Cambridge School of Clinical Medicine, Institute of Metabolic Science, Cambridge Biomedical Campus, Cambridge, United Kingdom, 2 Faculty of health, School of Kinesiology & Health Science, Keele Campus, Toronto, Canada, 3 Centre on Population Approaches for Non-Communicable Disease Prevention, Nuffield Department of Population Health, University of Oxford, Headington, Oxford, United Kingdom, 4 Warwick Medical School, University of Warwick, Division of Health Sciences, Coventry, United Kingdom, 5 Population Health Innovation Lab, Department of Public Health, Environment and Society, London School of Hygiene and Tropical Medicine, London, United Kingdom, 6 National Institute of Health Research Oxford Biomedical Research Centre, Oxford University Hospitals NHS Foundation Trust, Headington, Oxford, United Kingdom, 7 Department of Social and Policy Sciences, University of Bath, Claverton Down, Bath, United Kingdom, 8 College of Medicine and Health, University of Exeter, Medical School Building, St Luke's Campus, Exeter, United Kingdom

* nina.rogers@mrc-epid.cam.ac.uk

## Abstract

### Background

Sugar-sweetened beverage (SSB) consumption is positively associated with obesity, type 2 diabetes and cardiovascular disease. The World Health Organization recommends that member states implement effective taxes on SSBs to reduce consumption. The UK Soft Drinks Industry Levy (SDIL) is a two tiered tax, announced in March 2016 and implemented in April 2018. Drinks with ≥8g of sugar per 100ml (higher levy tier) are taxed at £0.24 per litre, drinks with ≥5-<8g of sugar per 100ml (lower levy tier) are taxed at £0.18 per litre, and drinks with <5g sugar per 100ml (no levy) are not taxed. Milk-based drinks, pure fruit juices, drinks sold as powder and drinks with >1.2% alcohol by volume are exempt. We aimed to determine whether the announcement of the SDIL was associated with anticipatory changes in purchases of soft drinks prior to implementation of the SDIL in April 2018. We explored differences in the volume of, and amount of sugar in, household purchases of drinks in each levy tier at two years post-announcement.

**Data Availability Statement:** The statistical code for the analyses are available from https://github.com/MRC-Epid/SDILEvaluation. Kantar Worldpanel

data are not publicly available but can be purchased from Kantar Worldpanel (http://www.kantarworldpanel.com). The authors are not legally permitted to share the data used for this study, but interested parties may contact Kantar WorldPanel representative Sean Cannon to inquire about accessing this proprietary data (Sean.Cannon@kantar.com).

**Funding:** DP, TLP, OM, MW and JA were funded by the Centre for Diet and Activity Research (CEDAR) (https://www.cedar.iph.cam.ac.uk/), a UKCRC Public Health Research Centre of Excellence. Funding from the British Heart Foundation, Cancer Research UK, the Economic and Social Research Council, the Medical Research Council, the National Institute for Health Research, and the Wellcome Trust, under the auspices of the UK Clinical Research Collaboration, is gratefully acknowledged. This work was supported by the Medical Research Council grant numbers MC_UU_12015/6 and MC/UU/00006/7, and by the NIHR Public Health Research programme (project No 16/130/01) (https://www.nihr.ac.uk/explore-nihr/funding-programmes/public-health-research.htm). The funders had no role in study design, data collection and analysis, decision to publish, or preparation of the manuscript.

**Competing interests:** I have read the journal's policy and the authors of this manuscript have the following competing interests: AB is a co-applicant on the National Institute for Health Research grant number 16/130/01, evaluating the impact of the UK Soft Drink Industry Levy, and is a past member of the UK Health Forum and current member of the Faculty of Public Health. Both of these organisations have a position statement supporting a soft drink tax. MR is Chair of Sustain: The alliance for better food and farming, a UK-based NGO. Sustain has advocated for the introduction and development of the SDIL. MW is director of the NIHR Public Health Research Funding programme; OM is currently on secondment at the UK Department of Health and Social Care and previously worked with Public Health England. JA is an academic editor for *PLOS Medicine*. Other than this, there was no support from any organisation for the submitted work other than that described above; no financial relationships with any organizations that might have an interest in the submitted work in the previous 3 years; and no other relationships or activities that could appear to have influenced the submitted work.

**Abbreviations:** CI, confidence interval; ITS, interrupted time series; KWP, Kantar Worldpanel; NHS, National Health Service; SDIL, Soft Drinks Industry Levy; SSB, sugar-sweetened beverage;

## Methods and findings

We used controlled interrupted time series to compare observed changes associated with the announcement of the SDIL to the counterfactual scenario of no announcement. We used data from Kantar Worldpanel, a commercial household purchasing panel with approximately 30,000 British members that includes linked nutritional data on purchases. We conducted separate analyses for drinks liable for the SDIL in the higher, lower and no levy tiers, and all liable and exempt soft drinks combined, controlling with household purchase volumes of toiletries.

At two years post-announcement against a backdrop of marked ongoing declines, there was a 41.3ml (95%CI 19.0 to 63.7ml) increase in volume of and a 5.1g (95%CI 2.0 to 8.1g) increase in sugar from, purchases of higher levy tier drinks compared to the counterfactual of no announcement (equivalent to 9.1% and 10.2% respectively). In contrast, a reversal of the existing upward trend in volume of, and amount of sugar in, purchases of lower levy tier drinks was seen. These changes led to a 68.1ml (95% CI: 54.9 to 81.1) reduction in volume and 4.4g (95% CI: 2.6 to 6.3) reduction in sugar purchased in these drinks per household per week compared to the counterfactual—a 38% reduction in both cases.

There was a 10% increase in volume of and 69% increase in sugar in household purchases of no levy drinks. At two years post-announcement, these changes led to a 165.5 ml (95%CI 100.1 to 230.9 ml) and 5.7 g (95% CI 4.0 to 7.3) increase in volume and sugar purchased in these drinks per household per week respectively, compared to the counterfactual. There was no evidence that total volume of purchases of all drinks combined was different from the counterfactual, but there was evidence of a small increase in sugar purchased from all drinks.

This is an observational study and changes other than the SDIL may have been responsible for the results reported. Purchases consumed outside of the home were not accounted for.

## Conclusions

The announcement of the UK SDIL was associated with reductions in volume and sugar purchased in lower levy tier drinks before implementation. These were offset by increases in purchasing of higher-levy and no levy drinks. These findings may reflect reformulation of drinks from the lower to no levy tier with removal of some, but not, all sugar, alongside changes in consumer attitudes, beliefs and purchasing behaviours.

## Trial registration

ISRCTN Registry ISRCTN18042742.

---

## Author summary

### Why was this study done?

Consumption of sugar-sweetened beverages (SSBs) is associated with dental caries, obesity, type 2 diabetes, and cardiovascular disease.

STROBE, Strengthening the Reporting of
Observational Studies in Epidemiology.

In March 2016, the UK government announced a tax on SSB manufacturers and produc-
ers, called the Soft Drinks Industry Levy (SDIL).

The SDIL has a higher tier of £0.24 per litre on drinks containing ≥8 g of sugar per 100 ml
and a lower tier of £0.18 per litre on drinks containing 5 to <8 g of sugar per 100 ml.

The SDIL was announced 2 years before it was implemented to allow manufacturers time
to reformulate products and so may have led to changes in purchases of SSBs even before
it was implemented.

### What did the researchers do and find?

We analysed purchases of drinks in the two levy tiers and other non-taxed drinks catego-
ries from 2014 to 2018 covering 2 years before the SDIL was announced and the 2 years
after it was announced, but before it was implemented.

We accounted for: the increase in purchases that occurred in the weeks leading up to
Christmas and Easter; the fall in purchases in the weeks after Christmas; and temperature,
as purchases tend to increase during warmer months.

We found that the purchased volume of and the amount of sugar in lower-tier drinks
reduced but that the purchased volume of and amount of sugar in the no-levy category
(containing drinks with <5 g of sugar per 100 ml) and the higher-tier category (contain-
ing drinks with ≥8 g of sugar per 100 ml) increased compared to the counterfactual of
what would have been expected without the announcement of the SDIL. The decrease in
lower-tier and increase in no-levy drinks may be because drinks from the taxed categories
had enough sugar removed to be exempt from the tax and therefore moved in to the no-
levy category.

The increase in household purchases of higher-tier drinks compared to the counterfactual
was against a backdrop of a strong existing downward trend in purchases of these drinks.

### What do these findings mean?

Taxes on SSBs are becoming more common throughout the world, but they vary in
design. Households started changing what they purchased in the 2-year period between
announcement and implementation of the SDIL because they were aware of the tax and
purchased different drinks, manufacturers removed the sugar in these drinks, or both.
Overall, the announcement of the SDIL was associated with a small increase in the total
amount of sugar purchased in drinks; further action, including implementation of the
SDIL, will be required to achieve public health benefit.

## Introduction

Sugar-sweetened beverage (SSB) consumption is positively associated with dental caries, total
energy intake, obesity, type 2 diabetes and cardiovascular disease [1–3]. Each additional daily
serving of SSBs consumed on a regular basis is associated with an 18% increased risk of type 2
diabetes and a 17% increased risk of coronary heart disease [2,3]. The economic burden of this
is significant. Obesity cost the UK economy around £27 billion in 2015 [4], with direct costs to
the NHS of over £5 billion [5].

The World Health Organization recommends that member states implement effective taxes
on SSBs to reduce consumption [6,7]. A number of national and regional governments includ-
ing Mexico, France, and multiple US cities have introduced SSB taxes [8–14]. Although the

longer term impacts of these taxes upon obesity rates are yet to be observed, short and medium term investigations have reported a drop in both SSB purchasing and consumption [10,12,15–17]. Many of these taxes generate an associated increase in the price of SSBs [9,18,19], which may be responsible for impacts on purchasing and consumption. However, there may also be other mechanisms of effect, including signalling of the health risks associated with SSBs, changes in social norms, reduced portion sizes, and reformulation of drinks [20]. To date these alternative mechanisms have received less research attention.

On March 16[th] 2016 the UK government announced the Soft Drinks Industry Levy (SDIL). The SDIL was the first SSB tax explicitly designed to incentivise a reduction in the amount of sugar in SSBs through reformulation [21]. This is reflected in the two levy tiers: £0.24 per litre for drinks containing ≥8g total sugar per 100ml and £0.18 per litre for drinks containing ≥5g and <8g total sugar per 100ml. Drinks containing <5g total sugar per 100ml are not taxed. Drinks containing at least 75% milk or milk alternatives, low and no alcohol drinks marketed as direct replacements for alcoholic drinks with <1.2% alcohol by volume, no added sugar fruit juice, drinks sold as powders, alcoholic drinks with >1.2% alcohol by volume, infant formula, and drinks for special medical purposes are exempt from the SDIL irrespective of sugar content. Products from small manufacturers and producers with annual sales of <1M litres of liable drinks are also exempt [22]. A number of other countries, including Ireland [23] and South Africa [24], have recently introduced similar taxes based on sugar concentration.

The announcement of the SDIL included a stated implementation date of April 2018 [15], giving manufacturers two years to adapt (e.g. reformulate their products or introduce new ones) prior to implementation. The announcement received extensive media coverage [25] and, together with discussion surrounding the potential harms of SSBs, may have itself impacted purchasing and consumption via changes in attitudes and beliefs. Furthermore, manufacturers had begun to introduce reformulated and new lower sugar, products during the two year adaptation period. The availability of SDIL liable soft drinks on supermarket shelves fell 19.5% almost two years after the announcement of the SDIL [26]. Whilst it has been reported that sales of levy eligible soft drinks fell by 50% from 2015–18, leading to an overall reduction in the amount of sugar in purchased soft drinks of 30% [27], this before-after study was not able to distinguish the impact of the SDIL from other trends in soft drinks purchases.

In line with recent developments in the public health literature [28], our evaluation theorised the SDIL as a series of events (specifically the announcement and implementation of the levy and related responses from relevant actors) in a complex adaptive system and planned analyses to evaluate the impact of each event [29]. In this framing, the SDIL announcement forms an important early phase of the intervention that is related to, but distinct from, the implementation. The two years between announcement and implementation was intended to give soft drinks manufacturers and producers time to respond to the SDIL under the assumption that it would be implemented. During this period changes in the availability of soft drinks occurred [26] which appear to have specifically been in anticipation of the implementation. It is important to explore the impact of this preparatory stage as, were the levy to be repealed, these anticipatory responses may not be reversed. By studying the impact of the SDIL on household purchases of soft drinks following the announcement along with post-implementation changes and other work examining the impact on health outcomes, the wider food and drinks industry and economy, and integrating findings in our overall SDIL evaluation [30], we aim to build up a comprehensive picture of the impacts of the SDIL and the mechanisms through which those were achieved [30].

In this paper we aimed to determine if the announcement of the SDIL was associated with anticipatory changes in purchases of soft drinks prior to implementation of the SDIL in April

2018 [28]. We have explored differences in the volume of, or amount of sugar in, household purchases of drinks in each levy tier, exempt drinks categories, and confectionery at two years post-announcement. We used controlled interrupted time series (ITS) methods with toiletries included as a control category, which we hypothesised to be unaffected by the SDIL, to take account of underlying trends in household purchasing. This allows existing purchasing trends to be taken into account when comparing pre- to post-announcement data, and prediction of longer-term changes in purchasing, compared to the counterfactual scenario. We compared observed changes associated with the announcement of the SDIL to the counterfactual scenario in which the announcement did not take place. The protocol was published [29] and the study was registered [ISRCTN18042742][30].

## Methods

National-level policy interventions such as the SDIL are not amenable to evaluation using randomised controlled trials. Controlled ITS analysis offers a robust observational method that allows the impact of the announcement to be investigated by both examining immediate changes in purchases and trends in these over time in comparison to counterfactual scenarios [31]. The counterfactual is the trend that would have occurred if the SDIL was not announced and is estimated by extrapolating the pre-announcement trend. A controlled design, that uses a product category likely to be unaffected by the announcement of the SDIL (i.e. toiletries), takes account of underlying trends in overall household purchasing [32,33]. In this study our primary outcomes were differences in the volume of, and amount of sugar in, purchases of drinks in each levy tier and exempt categories per household per week compared to the counterfactual scenario of no announcement, at two years post-announcement. To assess whether households would consciously or sub-consciously maintain their sugar intake by switching from SSBs to alternate high sugar products [34,35], we also studied trends in purchased total pack weight and sugar content of confectionery.

This study is reported as per the Strengthening the Reporting of Observational Studies in Epidemiology STROBE guideline (S1 Checklist).

### Data source

We used data from a commercial household purchasing panel, with approximately 30,000 British members, that includes linked nutritional data on food and drink purchases (Kantar Worldpanel (KWP)) aggregated to the weekly level. This allows purchases of SSBs to be examined in detail over time and compares favourably to other measures of food purchases [36]. Households were recruited using quota sampling with quotas for region, household size, age of main shopper, occupation and number of children in the household. Household purchasing of all drinks (including alcoholic drinks), sugar confectionery and chocolate confectionery (referred to collectively here as 'confectionery'), and shampoo, conditioner and liquid soap (referred to collectively here as 'toiletries') recorded by KWP households between 3rd March 2014 and 25th March 2018 were included. We selected shampoo, conditioner and liquid soap from the wider category of all health and beauty products on the basis that they were not likely to be seasonally dependent (such as for sun cream), not likely to be impacted by changes in sugar consumption (such as toothpaste), not likely to be impacted by households composition (such as gender biased products like make-up), sold by volume (rather than units unlike, for example, soap bars) and purchased volumes were of a similar magnitude to drink purchases. Panel households record information by scanning the barcode of all purchases brought into the home. Panel members receive points, exchangeable for gift vouchers worth approximately

£100 per year, as an incentive for taking part, and report household demographic information annually.

Purchase data are sent electronically from participating households to KWP each week and linked to nutritional compositional data (including sugar content) collected on a rolling basis by KWP field workers, covering all products every six months. Thus, nutritional data associated with each product in KWP changes over time in response to changes on product labels. Composition data for new products are collected when 20 purchases have been recorded within a three month period. KWP field workers visit supermarkets and photograph nutritional information panels identified by their barcode. Photographs are transcribed and linked via barcodes to purchasing records.

When a product cannot be found in any supermarket, nutritional information is substituted from elsewhere. Ideally, data from a different sized product in the same brand is used, (e.g. nutritional information from a 500ml bottle for a 330ml can of a specific cola brand). If no alternative within the same brand exists, the mean nutritional data of all similar products is imputed (e.g. all cola drinks). Where a product included a mix of imputed and observed sugar content values over the study period, we replaced imputed values by the last previously observed valued. As nutritional information panels are not generally displayed on alcoholic drinks in the UK, we did not study sugar in purchases of alcoholic drinks. Products in other categories that contained only imputed sugar values were excluded. This included all products in the skimmed milk category. We checked the validity of the remaining nutritional data and found that it was highly correlated with contemporaneous nutritional data on supermarket websites (see S5 Text and S3 Fig) and that the imputed products were spread evenly across drinks categories.

Households that record five or fewer purchases per week are excluded by KWP along with households whose adjusted weekly spending does not meet an undisclosed proprietary minimum value. KWP applies weights to purchases to adjust for households excluded due to minimum purchase or spending thresholds, and to maintain the representativeness of the panel. These weights were used in all analyses and ensure that the panel, and all purchases within it, are considered representative of all British households and purchases. In particular, our data represent mean purchases per household per week across all British households including non-purchasing households, rather than across all British households that purchased a particular product, or group of products.

## Drink categories

Drinks liable for the SDIL were classified into three categories: drinks containing $\geq$8g total sugar per 100ml (higher levy tier); drinks containing $\geq$5g to <8g total sugar per 100ml (lower levy tier) and drinks containing <5g total sugar per 100ml (no levy). Non-exempt drinks containing <5g total sugar per 100ml were sub-categorised into flavoured drinks containing >0g to <5g total sugar per 100ml, flavoured drinks containing 0g of sugar per 100ml, and bottled water. Exempt drinks were categorised as: alcoholic drinks containing >1.2% alcohol by volume and drinks with less alcohol by volume that are marketed as direct replacements for alcoholic drinks (collectively termed 'alcoholic drinks'); milk and milk based drinks containing at least 75% milk or milk alternatives (e.g. soy and almond drinks); no added sugar fruit juice; and drinks sold as powders (e.g. teas, coffees and hot chocolate). The SDIL includes further exemptions for infant formulas and foods for special medical purposes that were not examined [21].

## Analysis

**Main analysis.**   We conducted separate controlled ITS analyses for each drinks category and confectionery controlling for household purchase volumes of toiletries. We used 212 weekly time points from 3rd March 2014 to 25th March 2018 giving 107 pre- and 105 post-announcement weeks. The model is specified as follows:

$$Y_t = \beta_0 + \beta_1 T_t + \beta_2 X_t + \beta_3 X_t T_t + \beta_4 Z + \beta_5 Z T_t + \beta_6 Z X_t + \beta_7 Z X_t T_t + e_t$$

where $Y_t$ is the outcome (average weekly household purchases) at time $t$, $\beta_0$ to $\beta_3$ are the coefficients for the control group, and $\beta_4$ to $\beta_7$ represent the intervention group (drinks category or confectionery). $T_t$ represents the number of weeks since the first time point, a dummy variable is given by $X_t$ where 0 indicates the period prior to the SDIL announcement and 1 indicates the period after the announcement and $X_t T_t$ is the interaction of the announcement and the time since the start of the study allowing the trend following the announcement to be modelled. The intervention is included through $Z$, a dummy variable indicating drink or confectionery category or control category, $\beta_4$ gives the difference between the treatment and counterfactual before the announcement. The difference in the slope between the treatment and control groups before the announcement is given by $\beta_5$ and the difference in the level in the week immediately after the announcement is given by $\beta_6$, $\beta_7$ gives the difference between the slopes in the treatment and control groups after the announcement and $e_t$ is the variability at time $t$ not explained by the model [37]. No evidence of stationarity in each time series of volume and sugar was found using augmented Dickey-Fuller tests (both without and with trend). Dummy indicator variables determined to be statistically significant ($p<0.05$) were included as appropriate representing: the increase in purchases seen throughout December in the weeks before Christmas; the fall in purchases in the weeks immediately after Christmas; and the increase in confectionery purchases seen at Easter. To adjust for seasonality and temperature-related trends in drink consumption the average UK monthly temperature at each weekly time point was included [38]. Quadratic functions of trend $TX_t$ were included where they improved model fit—assessed using likelihood ratio tests. Autocorrelation between preceding time points was examined using Durbin-Watson tests and autocorrelation and partial-autocorrelation plots. An appropriate autocorrelation structure was determined and then compared to alternative models using likelihood ratio tests. Visual inspection of the data suggested no additional benefit would be gained from including polynomial terms.

Absolute and relative differences between observed post-announcement purchasing and the counterfactual scenario (assuming pre-announcement trends continued post-announcement) at 105 weeks ('2 years') post-announcement are presented, with 95% confidence intervals for the relative difference calculated using the delta method [39].

**Sensitivity analysis 1: Exemptions for small manufactures and producers.**   Soft drinks manufacturers and producers with annual sales of <1M litres of liable drinks are excluded from the levy. Relevant manufacturers are required to self-identify to Her Majesty's Revenue and Customs via their tax returns. A list of exempt manufacturers was not available to us, therefore, in the main analysis, products from all manufacturers were included. To estimate whether results were impacted by excluding smaller manufacturers, we estimated annual sales per manufacturer by summing purchases of liable drinks by manufacturer within each year. Thus, in this sensitivity analysis, we repeated the analysis described above firstly excluding liable products from manufacturers and producers with less than an average of 1M litres per year in our dataset. In addition, as the KWP data we used only captures purchases brought in to the home, it underestimates total sales. Therefore we performed a further set of analyses excluding manufacturers with less than, a more conservative average of, 0.5M litres per year.

**Sensitivity analysis 2: Combining drinks categories.** The SDIL does not apply to products such as fruit juices and milk-based drinks that may contain comparable amounts of sugar to SDIL liable products. To examine the extent to which the SDIL impacted upon the purchased volume and amount of sugar in all soft drinks, regardless of their SDIL liability, we also examined purchases of all non-alcoholic drinks combined. Controlled ITS analysis was carried out as above using all drinks categorised using the SDIL tier thresholds; as well as all drinks combined.

**Sensitivity analysis 3: Uncontrolled interrupted time series analysis.** A priori, we selected toiletries as a suitable control category for the reasons described above. It is possible, however, that a more appropriate control exists or that 'no category' is an appropriate control. We were not able to examine alternative controls but we are able to examine the impact of the selected control on the results we present. To this end we replicated the main analyses for drink and confectionery categories as described above with no control.

## Changes to protocol

A number of changes to the published protocol were made. First, rather than use data from two full years pre- to two full years post-announcement, we included data from 107 weeks pre- to 105 weeks post-announcement reflecting a small amount of additional data made available to us by KWP. Second, we did not include data on out-of-home purchases as indicated in the protocol. Whilst KWP does have a recently established panel capturing out of home purchases, data is only available on a subset of households and only from June 2015 and we did not feel this would provide a robust pre-intervention period. Third, we analysed purchasing at the weekly, rather than 4-weekly level reflecting an advantageous change in the data that KWP made available to us. Further analyses specified in the protocol [29] will be presented in future papers.

## Results

Of approximately 27 million purchases in the drinks, confectionery and toiletries categories over 212 weeks, 7% were for non-alcoholic drinks with only imputed sugar values and were excluded from the analyses. A further 0.5% contained a mix of imputed and observed values and were retained with last observed values carried forward.

An average of 22,265 households reported purchases in included categories each week. The characteristics of included households, including household size, remained consistent over the study period. Most panel households did not include children, were in managerial occupations (social grades AB or C1), and earned less than £40,000 per annum. Less than one fifth of chief income earners in included households had a degree-level education. The characteristics of included households, after weighting are compared to UK households as a whole in 2014–18 in S1 Text [40–43].

Table 1 presents the unadjusted mean purchases of drinks in each category, confectionery and toiletries per household per week pre- and post-announcement. Overall, mean volume of levy liable drinks purchased per household per week was 143.1ml lower in the post- versus pre-announcement period, with a corresponding 21.8g reduction in the amount of sugar purchased per week. This was primarily attributable to a reduction in purchasing of drinks in the higher levy tier.

Summaries of the controlled ITS models are shown in Tables 2 and 3. These tables document level and trend changes in the volume of, and amount of sugar in, purchases per household per week, and absolute and relative differences two years post-announcement, compared to the counterfactual. The level change is the difference between the model estimates and the

**Table 1. Unadjusted mean (sd) volume of, and amount of sugar in, purchased drinks and confectionery per household per week pre- and post-announcement of the Soft Drinks Industry Levy, March 2014 to March 2018.**

| Category | Mean (sd) volume (ml/g) | | Mean (sd) amount of sugar (g) | |
|---|---|---|---|---|
| | Pre-SDIL announcement | Post-SDIL announcement | Pre-SDIL announcement | Post-SDIL announcement |
| All soft drinks (i.e. excluding alcohol) | 7595.2 (295.3) | 7547.5 (466.1) | 336.7 (23.6) | 363.6 (17.1) |
| Liable drinks | | | | |
| Higher tier (≥8g sugar per 100ml) | 880.4 (128.1) | 680.3 (136.4) | 97.7 (14.1) | 75.5 (14.8) |
| Lower tier (≥5g–<8g sugar per 100ml) | 154.6 (32.2) | 147.0 (36.50) | 10.0 (2.15) | 9.69 (2.37) |
| No levy (<5g sugar per 100ml) | 1811.4 (168.9) | 1876.0 (215.8) | 11.8 (1.56) | 12.5 (2.72) |
| >0g to <5g sugar per 100ml | 784.9 (78.3) | 768.4 (92.2) | 11.8 (1.56) | 12.5 (2.72) |
| 0g sugar per 100ml | 1026.5 (104.5) | 1107.5 (132.1) | 0 | 0 |
| Bottled water | 590.9 (72.4) | 714.2 (90.6) | 0 | 0 |
| Exempt drinks | | | | |
| Alcoholic drinks | 1873.8 (380.4) | 1872.2 (455.7) | - | - |
| Milk and milk based drinks | 3546.4 (136.9) | 3540.4 (155.4) | 172.5 (6.60) | 171.8 (7.59) |
| No added sugar fruit juices | 516.6 (29.2) | 501.9 (43.6) | 50.9 (3.0) | 48.6 (4.2) |
| Drinks sold as powders (g) | 94.9 (11.7) | 87.7 (10.6) | 20.6 (3.17) | 18.5 (3.12) |
| Confectionery (g) | 308.4 (91.5) | 303.2 (92.7) | 173.3 (51.2) | 170.1 (52.1) |
| Toiletries | 122.6 (8.07) | 119.7 (8.2) | - | - |

SDIL: soft drinks industry levy

counterfactual at the first week after the SDIL announcement controlling for the underlying trends in household purchases through purchases of toiletries. The trend change is the mean change in the slope of purchases following the announcement. The absolute and relative differences represent the difference between the counterfactual and the model estimates in the final week of the study. Results are displayed graphically in Figs 1 and 2 for levy eligible drinks and

**Table 2. Adjusted change in mean volume of drinks (ml) and confectionery (g) purchased per household per week (95% CI) (level) and adjusted change per week (trend) post-announcement of the Soft Drinks Industry Levy including toiletries as a control condition, with absolute and relative differences in purchased volume at two-years post-announcement.**

| Category | Level change (ml/g) | Trend change (ml/g per week) | Change at 2 years post-announcement | |
|---|---|---|---|---|
| | | | Absolute change (ml/g) | Relative change (%) |
| All drinks | 56.5 (-143.6, 256.6) | -0.5 (-3.7, 2.7) | 4.8 (-103.5, 113.1) | 0.07 (-1.4, 1.6) |
| Liable drinks | | | | |
| Higher tier (≥8g sugar per 100ml) | **51.3 (10.1, 92.5)** | -0.1 (-0.8, 0.6) | **41.3 (19.0, 63.7)** | **9.1 (4.2, 14.0)** |
| Lower tier (≥5g–<8g sugar per 100ml) | 0.2 (-0.5, 0.8) | **-0.6 (-1.1, -0.2)** | **-68.1 (-81.1, -54.9)** | **-38.4 (-45.8, -31.0)** |
| No levy (<5g sugar per 100ml) | -13.8 (-132.7, 105.1) | 1.7 (-0.4, 3.7) | **165.5 (100.1, 230.9)** | **10.0 (6.0, 13.9)** |
| Drinks with >0g–<5g sugar per 100ml | -25.7 (-75.9, 24.5) | **1.2 (0.4, 2.0)** | **98.7 (71.6, 125.8)** | **15.6 (11.3, 19.8)** |
| Drinks with 0g sugar per 100ml | 9.8 (-64.3, 83.9) | 0.5 (-0.8, 1.8) | **64.7 (23.4, 106.1)** | **6.3 (2.3, 10.3)** |
| Bottled water | 16.0 (-36.7, 68.7) | -0.5 (-1.5, 0.4) | **-39.8 (-69.0, -10.6)** | **-5.5 (-9.5, -1.5)** |
| Exempt drinks | | | | |
| Alcoholic drinks | -10.0 (-49.2, 29.2) | -0.09 (-0.5, 0.4) | -16.8 (21.7, -55.3) | -1.0 (-3.2, 1.2) |
| Milk and milk based drinks | -13.2 (-114.5, 88.1) | -1.3 (-3.5, 0.9) | **-153.0 (-88.6, -217.4)** | **-4.1 (-5.8, -2.4)** |
| No added sugar fruit juices | -5.0 (-33.2, 23.2) | 0.2 (-0.4, 0.8) | 14.1 (32.0, -3.8) | 3.0 (-0.8, 6.8) |
| Drinks sold as a powder (g) | -2.0 (-9.0, 4.0) | -0.06 (-0.2, 0.04) | **-7.0 (-3.7, -10.2)** | **-6.9 (-10.1, -3.6)** |
| Confectionery (g) | -6.9 (-48.3, 34.4) | -0.06 (-0.9, 0.8) | -15.3 (12.2, -42.7) | -6.7 (-18.7, 5.3) |

Estimates statistically significant at the p<0.05 level are highlighted in **bold**

**Table 3. Adjusted change in mean sugar in drinks and confectionery purchased per household per week (95% CI) (level) and adjusted change per week (trend) post-announcement of the Soft Drinks Industry Levy including toiletries as a control condition, with absolute and relative differences in purchased volume at two-years post-announcement.**

| Category | Level change (g) | Trend change (g per week) | Change at 2 years after the post-announcement | |
|---|---|---|---|---|
| | | | Absolute change (g) | Relative change (%) |
| All drinks | 5.9 (-1.6, 13.4) | -0.01 (-0.1, 0.1) | **5.3 (1.2, 9.4)** | **1.7 (0.4, 3.0)** |
| Liable drinks | | | | |
| Higher tier (≥8g sugar per 100ml) | **7.4 (1.8, 13.0)** | -0.03 (-0.1, 0.05) | **5.1 (2.0, 8.1)** | **10.2 (4.0, 16.4)** |
| Lower tier (≥5g–<8g sugar per 100ml) | 1.8 (-1.4, 4.9) | **-0.07 (-0.1, -0.03)** | **-4.4 (-6.3, -2.6)** | **-38.2 (-53.6, -22.8)** |
| No levy (<5g sugar per 100ml) | 0.2 (-2.7, 3.0) | **0.04 (0.001, 0.08)** | **5.7 (4.0, 7.3)** | **68.5 (48.6, 88.4)** |
| Drinks with >0g–<5g sugar per 100ml | 0.2 (-2.7, 3.0) | **0.04 (0.001, 0.08)** | **5.7 (4.0, 7.3)** | **68.5 (48.6, 88.4)** |
| Exempt drinks | | | | |
| Milk and milk based drinks | 3.7 (-1.4, 8.9) | **-0.08 (-0.2, -0.002)** | **-3.8 (-6.6, -1.0)** | **-2.1 (-3.7, -0.5)** |
| No added sugar fruit juices | 1.1 (-3.1, 5.3) | 0.002 (-0.06, 0.06) | **2.6 (0.3, 4.8)** | **5.6 (0.6, 10.7)** |
| Drinks sold as a powder (g) | 0.8 (-2.7, 4.2) | -0.01 (-0.07, 0.05) | 0.6 (-1.3, 2.5) | 3.0 (-6.3, 12.3) |
| Confectionery | -6.9 (-48.3, 34.4) | -0.06 (-0.9, 0.8) | -15.2 (-42.1, 11.7) | -6.6 (-18.4, 5.1) |

Estimates statistically significant at the p<0.05 level are highlighted in **bold**; drinks with 0g sugar per 100ml and bottled water are excluded as they contain no sugar; alcoholic drinks are excluded as no information on sugar content was available

confectionery (and in S1 and S2 Figs for sub-categories within the no levy tier and other drinks categories). The thin blue lines (counterfactual) display a continuation of the observed pre-announcement trend controlled for changes captured by toiletries. The impact of the announcement of the SDIL upon purchasing can be observed by comparing the thick blue lines with shadows (modelled observed data) to the thin blue lines (counterfactual). Each combination of drinks category or confectionery with control was estimated and adjusted for autocorrelation distinctly; as a result the plotted toiletries estimates may vary between combinations.

In the two years pre-announcement, there was a marked decline in household purchase volume of drinks in the higher levy tier and some indication of a small increase in purchasing of drinks in the lower levy tier. The former was reflected in a simultaneous decrease in total sugar purchased from higher levy tier drinks in the 2 years preannouncement.

The announcement of the SDIL was associated with a significant level increase in household weekly purchase volume (p-values <0.05) of drinks in the higher levy tier. However, there was no change in trend of either volume or sugar. At two years post-announcement there was an increase in volume of drinks purchased in the higher levy tier (41.3ml (95% confidence intervals 19.0 to 63.7ml) per household per week), equivalent to 9.1% (95%CI 4.2 to 14.0)) and amount of sugar (5.1 g (95% CI 2.0 to 8.1g) purchased from these drinks, equivalent to 10.2% (95% CI 4.0 to 16.4%)) compared to the counterfactual of no announcement.

In contrast, the SDIL announcement was associated with a reversal of the existing upward trend in volume of, and amount of sugar in, household purchases of drinks in the lower levy tier. These changes led to a 38.4% (95% CI: 31.0 to 45.8) fall in the volume of, and a 38.2% (95% CI: 22.8 to 53.6) fall in the amount of sugar from drinks in, purchases of lower levy tier drinks per household per week at two years post-announcement (equivalent to a 68.1ml (95% CI: 54.9 to 81.1) reduction in volume and 4.4g (95% CI: 2.6 to 6.3) reduction in sugar purchased in these drinks per household per week) compared to the counterfactual.

Whilst the announcement of the SDIL was not associated with a level or trend change in volume of household purchases of no levy drinks, there was an increase in the trend in the amount of sugar in household purchases of these drinks. At two years post-announcement,

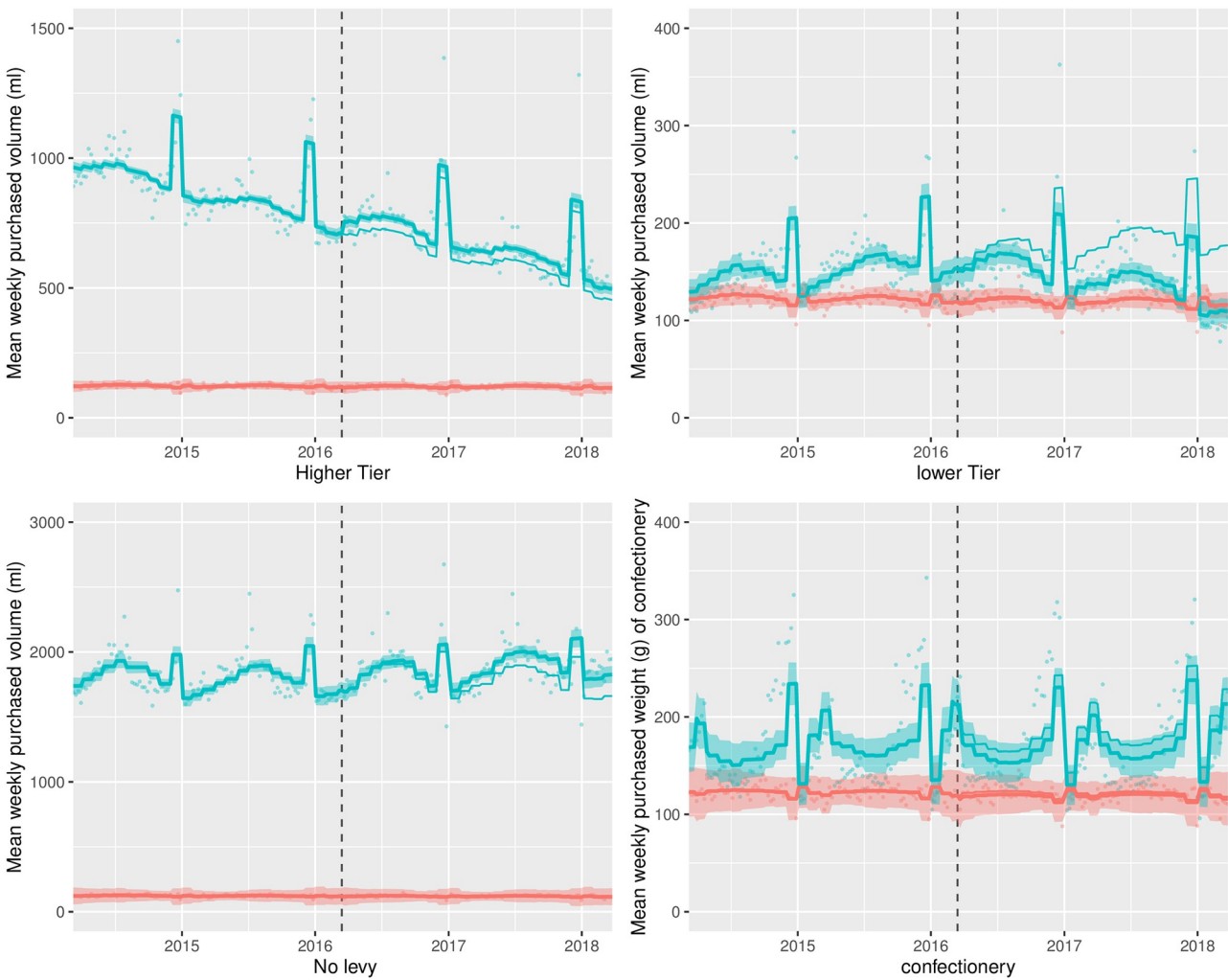

**Fig 1. Observed and modelled volume (ml) of drinks liable to the Soft Drinks Industry Levy and weight of confectionery (g) purchased per household per week, March 2014–March 2018.** Notes. Points are observed data, thick lines (with shadows) are modelled data (and 95% confidence intervals); thin lines (without shadows) are the counterfactual had the announcement not happened; blue points and lines are drinks/confectionery; red points and lines are the control category of toiletries; the vertical dashed line indicates the point of announcement of the SDIL; Y-axes vary in scale between panels to maximise the resolution of figures; modelled purchases include all adjustments as described in the methods section.

these changes led to a 68.5% (95% CI: 48.6 to 88.4) increase in the amount of sugar in purchases of no levy drinks per household per week (equivalent to a 5.7g (95% CI: 4.0 to 7.3) increase in sugar purchased in these drinks per household per week) compared to the counterfactual. At 2 years post-announcement, there was also a 10.0% (95% CI 6.0 to 13.9%) increase in volume of no-levy drinks purchased (equivalent to a 165.5ml (95% CI 100.1 to 230.9ml) increase). These increases in volume of, and sugar purchased from, no-levy drinks appeared to be driven by increases in non-water drinks (there was evidence of a decrease in volume of water purchased). The difference between the fall in sugar purchased from lower tier drinks and the increase in sugar purchased from no levy drinks may be due to the removal of some sugar by manufacturers meaning drinks shifted from the low levy tier to the no levy tier.

There was no evidence that the announcement of the SDIL was associated with changes in weight of, or amount of sugar in, confectionery purchased per household per week. The same

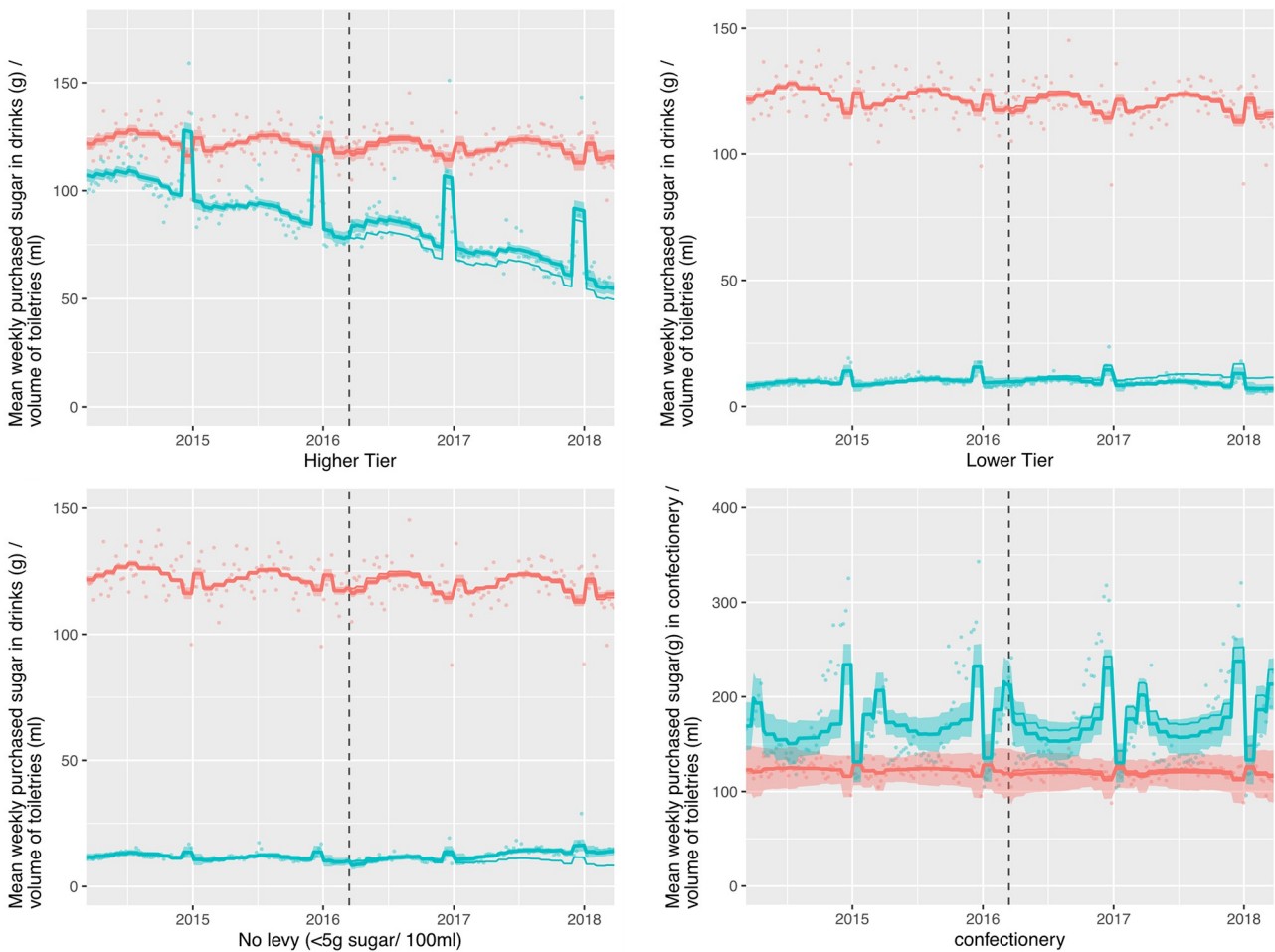

**Fig 2. Observed and modelled amount of sugar (g) in drinks liable to the Soft Drinks Industry Levy and confectionery (g) purchased per household per week, March 2014–March 2018.** Notes. Points are observed data, thick lines (with shadows) are modelled data (and 95% confidence intervals); thin lines (without shadows) are the counterfactual had the announcement not happened; blue points and lines are drinks/confectionary; red points and lines are the control category of toiletries; the vertical dashed line indicates the point of announcement of the SDIL; Y-axes vary in scale between panels to maximise the resolution of figures; modelled purchases include all adjustments as described in the methods section.

was true of alcoholic drinks. There were some small statistically significant changes in volume of and sugar purchased from exempt drink categories.

When all non-alcoholic drinks were combined, irrespective of levy eligibility or sugar content, at 2 years post-announcement, there was no change in volume purchased but a small increase in sugar purchased in drinks of 5.3g (95%CI 1.2 to 9.4g), equivalent to 1.7% (95% CI 0.4 to 3.0%) compared to the counterfactual.

Excluding manufacturers with sales of <1M litres or <0.5M litres per year (sensitivity analysis 1) led to very similar results (see S2 Text). When the SDIL thresholds of ≥8g, and >5g to <8g sugar per 100 ml were applied to group all drinks, including those exempt from the levy (sensitivity analysis 2), effect sizes seen in the main analysis were attenuated, but the direction and significance of effects remained unchanged (see S3 Text). When drinks from all of the non-alcoholic drinks categories were combined there was no evidence of a change in the volume of household drinks purchases as a result of the announcement of the levy at two years post-announcement but there was a small increase in sugar purchased from drinks of 5.3g

(95%CI: 1.2 to 9.4) or 1.7% (95%CI: 0.4 to 3.0) (see S3 Text) compared to the counterfactual. Overall, results were similar in both the models with and without toiletries acting as a control condition (see S4 Text).

## Discussion

### Summary of findings

We theorised that the announcement of the SDIL might lead to anticipatory changes, both by industry (as intended by government) and by consumers (via changes in consumer awareness, attitudes or beliefs). Two years after the announcement, immediately prior to implementation of the SDIL when all drinks were combined, irrespective of levy eligibility or sugar content and compared to the counterfactual of no announcement, we found no statistically significant change in the volume of all purchased soft drinks combined and a small increase in sugar purchased from these drinks of 1.7% or 5.3g per household per week, indicating that additional action (including implementation of the SDIL) is required to achieve a positive public health impact.

When we disaggregated levy-eligible drinks by category, we found evidence of a 9% increase in volume of, and a 10% increase in the amount of sugar in, household purchases of higher levy tier drinks compared to the counterfactual. This was against a backdrop of a substantial downward trend in purchasing of these drinks. Alongside, we found evidence of a 38% decrease in both the volume of, and the amount of sugar in, household purchases of lower levy tier drinks (equivalent to a reduction of 68ml and 4g of sugar in these drinks per household per week). There was also evidence of a 10% increase in the volume of household purchases of no levy drinks with <5g sugar per 100ml. The reduction in sugar purchased from lower levy tier drinks was more than offset by a 69% increase in the amount of sugar purchased in no levy tier drinks (equivalent to an increase of 6g of sugar in these drinks per household per week). We found no changes in purchasing of confectionery or alcoholic drinks associated with the announcement compared to the counterfactual scenario, suggesting that substitution to these categories did not occur. These results are consistent with either or both reformulation or introduction of new products, as well as consumer changes in purchasing. These findings help inform political discussions about rescinding sugar taxes (which has happened in, for example, Catalonia, Denmark and Cook County, Illinois, and is not therefore merely a theoretical concern). It should also help countries implementing similar taxes to understand the timeline they might expect for effects to occur (and hence when evaluations should take place and when any impact on health outcomes may be observed).

### Comparison of findings to previous research

Few previous studies have specifically explored the effects of the announcement of an SSB tax. In 2014, Chile changed the taxing structure of SSBs. A previous 13% *ad valorem* tax was reduced to 10% on drinks with <6.25g sugar per 100ml and increased to 18% for drinks with ≥6.25g sugar per 100ml. Drinks with no sugar, colouring or flavouring remained untaxed. In line with the current findings, announcement of the tax six months prior to implementation was associated with an anticipatory decline in purchasing of drinks in the low, but not high, tax group [27,40].

A recent ITS of drinks available in supermarkets, rather than drinks purchased, in the UK found that the announcement of the SDIL was associated with a 20% drop in the proportion of levy eligible drinks containing greater than 5g of sugar per 100ml (the minimum sugar threshold for the lower levy tier) [26]. Scarborough et al. also found evidence of 'strategic reformulation' with a new peak in the distribution of sugar content in drinks just below 5g per 100ml

that was not previously evidenced. This is consistent with our finding of reductions in purchasing of lower levy tier drinks and increases in the sugar purchased from no levy tier drinks (of 69%) that are proportionally much higher than the increase in volume purchased seen (of 10%).

A simple before-after analysis found that sales of all soft drinks increased by 5% from 2015–18, but total sugar from all soft drinks sold decreased by 30% [27]. In our unadjusted analysis, we found a drop of 5% in volume and 18% in sugar, which are in similar ranges. These differences in findings are likely attributable to methodological differences.

The 2018 annual report of the British Soft Drinks Association describes annual sales from 2012–17 of drinks in a range of categories [44] (data source not specified). According to these figures, mean annual sales of bottled water increased by 16% between 2014–15 and 2016–17 whilst those of no added sugar fruit juice decreased by 6%. Comparable figures from our unadjusted analyses (Table 1) are 21% and 3%. The report does not provide annual sales for the other categories used here.

Data in a report by Public Health England exploring effects of their sugar reduction strategy on purchases in 2015 (before announcement of the SDIL) and 2017 (after announcement but before implementation) report a decrease in sales of higher and lower levy tier drinks, but an increase in sales of no levy drinks [45]. We find the same direction of effect in our unadjusted analyses (Table 1).

## Interpretation of findings

We found that the announcement of the SDIL was associated with a marked reduction in the volume of, and sugar in, drinks purchased in the lower levy tier, which accelerated as the date of levy implementation approached. Alongside, we saw an increase in the amount of sugar purchased in no levy drinks, which also accelerated as the date of levy implementation approached. We hypothesise that these results reflect reformulation of many drinks to just below the maximum sugar content for the no levy tier (<5g of sugar per 100ml) and that this led to an overall increase in the average sugar content of drinks in this category. Prior to the levy announcement, this category was largely populated with zero sugar drinks, but after implementation a substantial new group of drinks with between 4.5g and 4.9g of sugar was seen [26].

We are not able to disentangle potential mechanisms of the changes in household purchasing associated with announcement of the SDIL reported here. Alongside reformulation, media coverage of the announcement and adaptive behaviours by industry, such as changes in marketing strategies, may have had an impact on purchasing by heightening both salience of SSBs and awareness of the health harms of SSBs. For example, in another study, Mexicans who were aware of their SSB tax were more likely to report a recent decrease in consumption [20].

We found a 9% increase in purchased volume of, and a 10% increase in sugar purchased from, drinks in the high levy tier compared to the counterfactual. These changes did not accelerate over time and appear attributable to a small increase in purchasing at the time of the announcement. It is possible that the small increase immediately following the announcement reflected stockpiling [46] in anticipation of price or recipe changes. Alternatively, this may reflect 'psychological reactance' whereby restrictive policies perceived to limit freedom lead to resistance, efforts to regain that freedom, and hence strengthening or adoption of the behaviour that the policy seeks to limit [47]. Studies from the UK [48] and US [49] have demonstrated that some people, particularly those consuming more SSBs, have negative emotional reactions to SSB taxes consistent with reactance. Some evidence of psychological reactance was

similarly found in relation to the vote to implement an SSB tax in Berkeley, California [50]; but was not replicated in analyses of the impact of implementation [10].

There was a marked existing downward trend in purchasing of high levy tier drinks, with unadjusted purchased volume declining by 23% during the study period. The higher levy tier category is dominated by market leading cola drinks, many of which have been reported as unlikely to reformulate [51]. Consumers of these products tend to have strong brand loyalty and are more likely to consume large volumes [52]. Any reformulation of drinks in this category may have been limited to products with small market shares. It is also possible that the large existing downward trend in purchasing represents the fastest that the market and consumers are able to change, with no additional impacts feasible from the announcement of the SDIL.

Potential unintended consequences of the SDIL include substitutions to other less healthful categories such as confectionery and alcoholic drinks. We had initially hypothesised that households would maintain sugar levels by switching from SSBs to confectionery [34,35,53]. However, we did not find evidence of this with no statistically significant changes in household purchases of confectionery observed. We also found no evidence that the SDIL announcement was associated with changes in purchases of alcohol compared to the counterfactual scenario. Changes in purchasing of confectionery and alcohol may still follow SDIL implementation and this should be monitored.

## Strengths and weaknesses of methods

This work was carried out using data from a large, national, household purchase panel. However, the KWP data used only captures purchases brought home. Whilst KWP has recently established a smaller 'out of home' panel, data are not available prior to June 2015 and are only available from a smaller sub-panel. It is possible that out of home purchasers responded differently to the announcement of the SDIL to those brought in to the home. However, the unadjusted changes in bottled water and no added fruit juice reported here compare favourably to similar data on the full UK market, indicating that our results may be generalizable to all purchasing.

Household purchases do not necessarily equate to individual consumption as drinks purchased may be wasted or shared unevenly within households. Nevertheless purchases brought into the home as captured by KWP appear to provide a reasonably accurate estimate of consumption [54]. An alternative approach would have been longitudinal analyses of individual household data. However, KWP is a consistently shifting panel of households who join and leave, or are temporarily dropped due to poor quality data. Further, due to data cost, we only have data on the categories reported here—drinks, toiletries and confectionery. Thus, where a household recorded purchases of drinks in earlier but not later weeks, we were unable to distinguish between the household having withdrawn from the panel, stopped purchasing soft drinks, or being excluded due to poor data quality, precluding an analysis at the individual household level.

We used KWP data on sugar content of purchased drinks. Nutritional data on existing products is checked by KWP every six months. This may lead to a lag between reformulation and changes in KWP data making it possible that we have underestimated changes in purchasing of sugar associated with reformulation. However, our validity checks of KWP sugar content data found no indication of systematic differences between KWP sugar content data and contemporaneous values listed on supermarket websites (see S3 Fig and S5 Text).

Attribution of effects in an ITS analysis is vulnerable to other interventions with the potential to impact on the outcomes of interest occurring at, or near, the same time. The

announcement of the SDIL was part of the UK Chancellor's 2016 budget speech. This contained other announcements that may have impacted on household purchases. The inclusion of a control category (toiletries) attempted to take any underlying changes in household purchasing into account. However, toiletries were selected before the data were purchased and thus they may be subject to trends that lessen (or magnify) the effects reported here, and a more appropriate group of products to function as a control may exist.

The SDIL can be seen as part of a wider dialogue surrounding sugar and SSB consumption in the UK that includes the Scientific Advisory Committee on Nutrition's report on carbohydrates and health, the government's ongoing childhood obesity plan (with chapters 1 and 2 published in 2016 and 2018 respectively) and sugar and calorie reduction strategies (published in 2017 and 2018 respectively) [4,55–58]. Publication of all of these, in addition to television documentaries such as "Jamie's Sugar Rush" [59] and "That Sugar Film" [60], may have played some part in the changes reported here.

We took a hypothesis-driven approach to focus our analysis on the point of announcement of the SDIL. A data-driven search for other inflexion points may have revealed different patterns, but we felt this would have been conceptually confusing and potentially difficult to interpret. Future work could explore this further.

Panel data can be limited by changes in panel composition over time. However, the demographic characteristics of the KWP panel remained similar over the study period. Furthermore, proprietary weightings provided by KWP and used throughout account for non-consumers and adjust for variations in panel composition. Resource constraints limited us from studying impacts across all food and drinks categories. Our analyses focus on the impact of the SDIL announcement on purchasing. Further work will be required to determine the impact of implementation on purchases and relevant health outcomes.

## Implications of findings

Our findings indicate that the announcement of the SDIL was associated with changes in purchasing of soft drinks in a number of categories that may be the result of product reformulation, changes in consumer awareness, attitudes and beliefs, or a combination of both. Overall, however, we found no change in total volume of household purchases of all soft drinks combined and a small increase in sugar purchased from all drinks combined, indicating that further action will be required beyond the announcement of the levy to achieve a positive public health impact—including the levy's actual implementation. The announcement of an SSB tax is not an isolated intervention that can likely be replicated in other contexts as the reaction to the announcement occurs on the assumption that the tax will be implemented. Therefore the announcement of an SSB tax on its own, whilst making it clear that it would not be implemented, is unlikely to produce the same findings as those reported here. Nevertheless, in line with recent developments in the public health literature [25], by theorising the SDIL as a series of events in a complex system, with the announcement as a key early event leading to changes in anticipation of implementation, our findings help to build a complete picture of the impacts of the SDIL, when these occurred and via what mechanisms.

## Conclusions

The announcement of the UK SDIL was associated with a 68.1ml decrease in the volume of household purchases of lower levy tier drinks containing ≥5-<8g sugar per 100ml per week compared to the counterfactual estimated from pre-announcement trends, equating to 4.4g less sugar per household per week. Against a background of substantial existing downward trends in volume of, and amount of sugar in, higher levy tier drinks with >8g sugar per 100ml

the announcement of the SDIL was associated with a 41.3 ml increase in the volume of household purchases of these drinks equating to 5.1g sugar per household per week compared to the counterfactual. There was also a 165.5ml increase in volume and 5.7g increase in sugar purchased from drinks in the no levy tier compared to the counterfactual. There was no evidence of substitution to alcoholic drinks or confectionery. These findings may reflect reformulation of drinks from the low levy to no levy tier with removal of some, but not all, sugar alongside changes in consumer attitudes, beliefs and purchasing behaviours. Any reformulation of drinks in the higher levy tier may have been limited to drinks with small market shares. Overall there was no change in total volume of household purchases of all soft drinks combined but there was a small increase in sugar purchased from these drinks, indicating that further action, including implementation of the SDIL, is required to achieve public health impact. Future work should determine the impacts of the implementation of the SDIL.

## Supporting information

**S1 Checklist. STROBE checklist of item that should be included in reports of cohort studies.** (STROBE: Strengthening the Reporting of Observational Studies in Epidemiology). (DOCX)

**S1 Fig. Observed and modelled volume (ml) of drinks purchased per household per week, March 2014–March 2018 (sub-categories within the no levy tier and other drinks categories).** Notes. Points are observed data, thick lines (with shadows) are modelled data (and 95% confidence intervals); thin lines (without shadows) are the counterfactual had the announcement not happened; blue points and lines are drinks; red points and lines are the control category of toiletries; the vertical dashed line indicates the point of announcement of the SDIL; Y-axes vary in scale between panels to maximise the resolution of figures; modelled purchases include all adjustments as described in the methods section. (TIF)

**S2 Fig. Observed and modelled amount of sugar (g) in drinks purchased per household per week, March 2014–March 2018 (sub-categories within the no levy tier and other drinks categories).** Notes. Points are observed data, thick lines (with shadows) are modelled data (and 95% confidence intervals); thin lines (without shadows) are the counterfactual had the announcement not happened; blue points and lines are drinks; red points and lines are the control category of toiletries; the vertical dashed line indicates the point of announcement of the SDIL; Y-axes vary in scale between panels to maximise the resolution of figures; modelled purchases include all adjustments as described in the methods section. (TIF)

**S3 Fig. Agreement plot between nutritional information reported by KWP and historic supermarket and manufacturer data from Archive.org.** (TIF)

**S1 Text. Demographic characteristics of Kantar Worldpanel households from March 2014–March 2018 (weighted).** (DOCX)

**S2 Text. Sensitivity analysis 1: Exemptions for small manufacturers and producers.** (DOCX)

**S3 Text. Sensitivity analysis 2: Combining drinks categories.** (DOCX)

**S4 Text. Sensitivity analysis 3: Uncontrolled interrupted time series analysis.**
(DOCX)

**S5 Text. Validity checks of data on sugar content of drinks.**
(DOCX)

## Acknowledgments

The views expressed are those of the authors and not necessarily those of the National Health Service, the NIHR, or the Department of Health and Social Care, UK. The funders had no role in study design, data collection and analysis, decision to publish, or preparation of the manuscript.

## Author Contributions

**Conceptualization:** Tarra L. Penney, Oliver Mytton, Adam Briggs, Steven Cummins, Mike Rayner, Harry Rutter, Peter Scarborough, Richard D. Smith, Martin White, Jean Adams.

**Data curation:** Nina T. Rogers, David Pell, Stephen J. Sharp.

**Formal analysis:** Nina T. Rogers, David Pell, Stephen J. Sharp.

**Funding acquisition:** Tarra L. Penney, Oliver Mytton, Adam Briggs, Steven Cummins, Mike Rayner, Harry Rutter, Peter Scarborough, Richard D. Smith, Martin White, Jean Adams.

**Investigation:** David Pell, Oliver Mytton, Jean Adams.

**Methodology:** Nina T. Rogers, David Pell.

**Project administration:** Tarra L. Penney.

**Supervision:** Jean Adams.

**Validation:** David Pell.

**Visualization:** Nina T. Rogers, David Pell, Oliver Mytton.

**Writing – original draft:** David Pell, Jean Adams.

**Writing – review & editing:** Nina T. Rogers, David Pell, Tarra L. Penney, Oliver Mytton, Adam Briggs, Steven Cummins, Mike Rayner, Harry Rutter, Peter Scarborough, Stephen J. Sharp, Richard D. Smith, Martin White, Jean Adams.

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
