## [Decision Letter · Decision Letter 0]

11 Feb 2020

Dear Dr. Pell,

Thank you very much for submitting your manuscript "Anticipatory changes in British household purchases of soft drinks associated with the announcement of the Soft Drinks Industry Levy: a controlled interrupted time series analysis [ISRCTN18042742]" (PMEDICINE-D-19-03444) for consideration at PLOS Medicine. 

[LINK]

In light of these reviews, I am afraid that we will not be able to accept the manuscript for publication in the journal in its current form, but we would like to consider a revised version that addresses the reviewers' and editors' comments. Obviously we cannot make any decision about publication until we have seen the revised manuscript and your response, and we plan to seek re-review by one or more of the reviewers. 

We expect to receive your revised manuscript by Mar 17 2020 11:59PM. Please email us (plosmedicine@plos.org) if you have any questions or concerns.

We look forward to receiving your revised manuscript. 

Sincerely,

Adya Misra, PhD

Senior Editor 

PLOS Medicine

plosmedicine.org

Title- Please remove [ISRCTN18042742] from the title

Abstract- please combine the methods and findings into one section. The last sentence of this section should be a limitation of your study design

Abstract-please provide the name of the household purchasing panel used

Abstract- perhaps this ought to be in the background section? “The SDIL is a two tiered tax, announced in March 2016 and implemented in April 2018.

Drinks with ≥8g of sugar per 100ml (higher levy tier) are taxed at £0.24 per litre, drinks

with ≥5-<8g of sugar per 100ml (lower levy tier) are taxed at £0.18 per litre and drinks

with <5g sugar per 100ml (no levy) are not taxed. Milk-based drinks, pure fruit juices,

drinks sold as powder and drinks with >1.2% alcohol by volume are exempt”. 

Data availability- please note that authors cannot be contacts for data requests. The code for analyses should be deposited in a repository and the details provided in the data availability statement. 

References- please provide the full stop after the square brackets and when multiple references are cited, these can be included within one set of brackets for example: [2,3] or [8-10]. 

Please discuss and cite all recently published articles related to this research in the Introduction/Discussion sections as appropriate. Specifically, Scarborough et al in PLOS Medicine and Bandy et al in BMC Medicine must be discussed to clarify why similar findings are being reported separately. 

STROBE checklist: please use section and paragraph numbers, rather than page numbers. Please add the following statement, or similar, to the Methods: "This study is reported as per the Strengthening the Reporting of Observational Studies in Epidemiology STROBE guideline (S1 Checklist)."

Comments from the reviewers:

Reviewer #1: 

The study estimates changes in volume and sugar purchased after the announcement of the two tiers levy for beverages. Although the topic is very relevant there are some methodological aspects that should be reviewed. One of the main limitations of the study is that the control group does not seem to follow the same pre-announcement trend neither the magnitudes look similar. I conclude this by looking at the graphs but the interrupted time series analyses provide actual tests by looking at the pre-announcement coefficients: the difference in the level (intercept) of the outcome variable between control and treatment categories prior to the announcement and the difference in the slope of the outcome variable between both categories prior to the announcement. These coefficients are not shown and are key to test for an appropriate control group. Also the 56% decrease in volume and sugar content seems very large, given that this was only the announcement, not the implementation of the tax.

In addition, the paper lacks a contextual framework to hypothesize on the potential changes over the announcement period. 

Introduction

Define in more detail what you mean by series of events (which ones) and complex adaptive systems. I don´t see any application of this in the methods.

Methods

Data- If the data representative of the British population?

Model- Interrupted times series analysis is a method applied to time series. I am not sure that the data is aggregated as time series because the authors are using a panel of households. This is confusing in the methods. Did the authors aggregated the data? At what level and why given the richness of using longitudinal data to explore heterogeneities.

To adjust for seasonality, it is not clear how often temperature is included, monthly, weekly? Any macroeconomic level variable that is associated with household purchases?

Control group: why toiletries? It is the only non-food and beverage items included?

Potential substitutions: I don´t see any justification as for why would confectionary be a substitute for SSB, no literature is cited. What was included in confectionery? How about untaxed beverages? It is very relevant to look at substitutions for no levy beverages.

The authors should test different inflexion points, right now they are assuming that the was an immediate change after announcement and change in the slope. This could have happened in different times in the post-announcement period. 

Results

What are AB or C1 classes?

Table 2 presents two estimations, what are the coefficients in columns 2 and 3 compared to the change at 2 years post-announcement?

How could there be an increase in sugar in the no levy category? How could reformulation lead to this increase, it makes little sense and the discussion does not provide much inside on these findings. 

Reviewer #2: Pell and colleagues present the findings of a quasi-experimental study evaluating the effects of the UK soft drinks industry levy on household purchasing of drinks. The modelling methods uses an interrupted time series analysis with a counterfactual control product of toiletries for comparison. They have concluded that there were reductions in purchasing due to the soft drinks levy lower levy tiered drinks, but generally this was offset by increases in purchasing on no levy drinks. The findings implicate that manufacturers could be reformulating to lower levy drinks to no levy drinks. The downstream health impacts of this will be interesting to see in future studies. 

This was a well-conducted and reported time series analysis used for policy evaluation. The methods are robust and the author explained their methods and approach clearly, which can be sometimes particularly challenging the reporting of ITS studies for a more general medical journal such as PLOS Medicine. The study itself is timely and novel and should have a generate some substantial impact as many policy-makers are waiting for this type of analysis to be conducted evaluating the UK SDIL. The panel dataset used to examine this was from a large number of household (30,000 members) and covered over 27 million purchases. Only a small amount of imputed data was necessary (0.5%) - suggesting the data was mostly complete. This paper should warrant publication barring a few minor issues to addressed first I've detailed below:

1) Abstract - primary outcome: give the unit of measurement for both volume and amount of sugar in these outcomes. 

2) Abstract - results: I it would make more sense to have a "-" sign in front the absolute reduction of volume and amount of sugar figures, to ensure consistency with the reported 95% CI

3) Introduction (and related methods): Selection of toiletries as the control - I don't disagree with the authors rationale on why this was selected as the control but in terms of the hypothesis that toiletries were sensitivity to disposable income? However, could the authors elaborate if they looked in their data whether this was in fact sensitive to disposable income? A simple exploration of the relationship between income and toiletry purchasing could confirm this as this looks to be one the key assumptions on control selection. 

4) Main analysis methods - Was there any testing or investigation of stationarity in both the experimental and counterfactual models? The design of having a counterfactual trend for toiletries for comparison partially controls for this issue, but assumes both products have similar levels of stationarity.

5) Main analysis methods - Did the authors consider any changes in household size of period of time their analysis period is over four year period - consumptions patterns change or behavioural patterns may changes due to households over time due to external influences (i.e. children)

6) Main analysis method - The authors considered the effects of period based purchasing (i.e. Christmas, Easter) using dummy indicator variables but I was wondering if the models were seasonally adjusted or included a seasonal term

7) Generating 95% CI with multivariate delta method: Explain the method is used to generate estimations of sampling variance (hence able to determine 95% CIs)

8) Sensitivity analyses - As the authors can appreciate, public health policy has differential treatment effects due to socioeconomics and levels of education. I noticed in the descriptive tables, there were figures on household income and social class. I was expecting to see a sensitivity analysis in this paper stratifying the effects of SIL by SES. What the key questions would have been very desirable to see if what effect SIL had between these stratums - and in fact would further enhance the overall findings. 

9) Again for consistency in results section - I would suggest having "-" in front when presenting reductions which would correspond to the reported 95% CIs. 

Reviewer #3: Anticipatory changes in British household purchases of soft drinks associated with the announcement of the Soft Drinks Industry Levy: a controlled interrupted time series analysis. Pell et al.

There has been much speculation about the time course of any changes in SSB consumption associated with the announcement and subsequent implementation of the SDIL.

This analysis is thus welcome.

I have very few suggestions. These are mostly about maximising clarity of what might otherwise be a potentially confusing mass of trend data.

ABSTRACT

The manuscript-based abstract correctly commences the Results section with the key finding.

"There was no evidence that volume of, or amount of sugar in, purchases of all drinks combined was different from the counterfactual."

This sentence appears to have been mistakenly moved to the end of the results in the web-based abstract. That should be corrected.

DISCUSSION

Para 1 is potentially confusing. I suggest that before embroiling the reader in the detail, the second sentence might usefully start with the key overall message. Something along the lines of :

"We theorised that the announcement of the SDIL might lead to anticipatory changes, both by industry (as intended by government) and by consumers (via changes in consumer awareness, attitudes or beliefs). 

 When all drinks were combined, we found no significant change in the volume of, or amount sugar in, purchased drinks."

The Concluding para ends: 

"Overall there was no change in total volume of, or sugar in, household purchases of all soft drinks combined indicating that further action, including implementation of the SDIL, is required to achieve public health 

 impact. Future work should determine the impacts of the implementation of the SDIL."

This is slightly disappointing.

It would be preferable to have one paper showing all the results together, to also see the changes post SDIL implementation.

Nil else.

[LINK]

---

## [Decision Letter · Decision Letter 1]

1 Jul 2020

Dear Dr. Pell,

Thank you very much for re-submitting your manuscript "Anticipatory changes in British household purchases of soft drinks associated with the announcement of the Soft Drinks Industry Levy: a controlled interrupted time series analysis [ISRCTN18042742]" (PMEDICINE-D-19-03444R1) for review by PLOS Medicine. I'm truly sorry for the extreme delay to your submission.

I have discussed the paper with my colleagues and the academic editor and it was also seen again by reviewers. I am pleased to say that provided the remaining editorial and production issues are dealt with we are planning to accept the paper for publication in the journal.

[LINK]

We look forward to receiving the revised manuscript by Jul 08 2020 11:59PM. 

Sincerely,

Adya Misra, PhD

Senior Editor 

PLOS Medicine

plosmedicine.org

Requests from Editors:

Title- please remove [ISRCTN18042742] from the title 

Competing interests- could you please add a sentence to note that Jean Adams is an Academic Editor for PLOS Medicine

References- please adapt to Vancouver style

Abstract- limitations should be made more explicit. For instance instead of "Only purchases brought into the home were included" you may say "out of home purchases could not be accounted for" or similar.

Page 5- please add a subheading “Author Summary” and remove the gray background here please

Author summary in the section “what do these findings mean” please can you tone down to remove causal language? For instance “Households started changing what they purchased in the two year period between announcement and implementation of the SDIL either because households were aware of the tax and purchased different drinks or more likely because manufacturers removed the sugar in these drinks”

Please provide p values where needed, especially where you mention “significance” for example page 18 and Table 2,3. These should be exact p values unless p<0.001

Comments from Reviewers:

Reviewer #2: I have re-reviewed the manuscript and found the authors addressed most of the reviewer comments adequately. 

I had one minor issue I was mulling over about in the additional supplemental analysis Tables E and F. The results are somewhat similar, with expected changes in effect sizes. The trends did reach statistical significance for purchase volume of powdered drinks and sugar in confectionary in the uncontrolled analysis. I agree that this would mean the use of controls was conservative choice but is there any type 2 error risk here with the choice of the control? The choice for toiletries as the control was largely pragmatic which I understand but I think a limitation is that the researcher could not access other household products to investigate this issue further. This should probably just be mentioned as a limitation of the analysis.

[LINK]

---

## [Editor Report · Decision Letter 2]

18 Sep 2020

Dear Dr Pell, 

On behalf of my colleagues and the academic editor, Dr. Barry M. Popkin, I am delighted to inform you that your manuscript entitled "Anticipatory changes in British household purchases of soft drinks associated with the announcement of the Soft Drinks Industry Levy: a controlled interrupted time series analysis" (PMEDICINE-D-19-03444R2) has been accepted for publication in PLOS Medicine. 

PRODUCTION PROCESS

PRESS

PROFILE INFORMATION

Thank you again for submitting the manuscript to PLOS Medicine. We look forward to publishing it. 

Best wishes, 

Adya Misra, PhD

Senior Editor 

PLOS Medicine

plosmedicine.org